# PAITS: Pretraining and Augmentation for Irregularly-Sampled Time Series

## Abstract

Real-world time series data that commonly reflect sequential human behavior are often uniquely irregularly sampled and sparse, with highly nonuniform sampling over time and entities. Yet, commonly-used pretraining and augmentation methods for time series are not specifically designed for such scenarios. In this paper, we present PAITS (Pretraining and Augmentation for Irregularly-sampled Time Series), a framework for identifying suitable pretraining strategies for sparse and irregularly sampled time series datasets. PAITS leverages a novel combination of NLP-inspired pretraining tasks and augmentations, and a random search to identify an effective strategy for a given dataset. We demonstrate that different datasets benefit from different pretraining choices. Compared with prior methods, our approach is better able to consistently improve pretraining across multiple datasets and domains. Our code is attached and will be publicly available.

## 1 Introduction

Time series data appear in many areas ranging from healthcare to retail, and play an important role in tasks such as forecasting to classification. Despite the abundance of time series data in a variety of fields, there is often a relative scarcity of labeled data, due to the fact that generating annotations often requires additional effort or expertise. In other domains, such as computer vision and natural language processing (NLP), large unlabeled datasets have motivated the use of unsupervised pre-training methods which have led to great improvements in downstream supervised tasks with smaller labeled datasets Chen et al. (2020); Devlin et al. (2018). While pretraining strategies for time series data have been relatively less explored, recent works have shown promise, for example using contrastive learning Eldele et al. (2021); Zhang et al. (2022); Yue et al. (2022). However, many such approaches have been developed for cases in which data appear in frequent and regular intervals, with repeating signals.

In real-world scenarios (such as medical data in healthcare settings), features may be collected at irregular intervals, and particularly for multivariate datasets, features may be collected at varying rates. When using a traditional approach of representing a time series as a matrix of features' values across regularly spaced time intervals (which we refer to as "discretization", as described by Shukla & Marlin (2020)), such irregularity can lead to challenges of extreme sparsity in the discretized data setting (i.e., high missingness for pre-defined intervals, as illustrated in Figure 1).

Some recent studies have instead cast time series data as a set of events, which are each characterized by a time, feature that was observed, its value at that time Horn et al. (2020); Tipirneni & Reddy (2022). Such a representation is more flexible, requiring minimal preprocessing, and avoids the need to explicitly represent "missingness," as the data only contains events that were observed (e.g., Figure 1). This data representation also has parallels to natural language: while NLP casts text as a sequence of words (tokens), this approach for time-series data represents a time series as a sequence of events (with an associated feature, time, and value), and thus we hypothesize that pretraining and augmentation approaches developed for NLP may be particularly advantageous for sparse and irregularly sampled time series data. In particular, we consider a forecasting task (previously explored by Tipirneni & Reddy (2022)) along with a sequence reconstruction task inspired by the pretraining strategies used in BERT Devlin et al. (2018).

**Representing irregularly sampled time series**

*Raw data:*

| Time | Feature | Value |
|------|---------|-------|
| 0:06 | Heart rate | 0.5 |
| 1:30 | Temperature | -0.3 |
| 2:22 | Sys. Blood Pressure | 0.8 |
| 3:05 | Heart rate | 0.1 |
| 3:31 | O$_2$ Saturation | 1.2 |
| 3:49 | Heart rate | -0.3 |
| ⋮ | | |

*Common discretized representation:*

| | $t_0$ | $t_1$ | $t_2$ | $t_3$ ⋯ |
|---|---|---|---|---|
| Heart rate | 0.5 | | | -0.1 |
| Temp. | | -0.3 | | |
| Sys. BP | | | 0.8 | |
| Glucose | | | | |
| O2 Sat. | | | | 1.2 |
| ⋮ | | | | ⋱ |

*Sequence representation in PAITS:*

| Time | 0.1 | 1.5 | 2.4 | 3.1 | 3.5 | 3.8 | ⋯ |
|------|-----|-----|-----|-----|-----|-----|---|
| Value | 0.5 | -0.3 | 0.8 | 0.1 | 1.2 | -0.3 | |
| Feat. | 1 | 2 | 3 | 1 | 5 | 1 | |

Figure 1: Illustration of discretized vs. sequence-based representation of irregularly sampled time series data.

We experimented with multiple pretraining tasks and related augmentations and found that there was not a one-size-fits-all approach that consistently worked best across multiple datasets. For example, when considering the same mortality prediction task in different datasets, we found that some benefited from the combination of two pretext tasks, whereas another was only helped by a single pretraining task; and notably, each dataset was best suited by differing augmentations. Because datasets benefit from different pretraining strategies, we present a framework, **P**retraining and **A**ugmentation for **I**rregularly-sampled **T**ime **S**eries (PAITS), to identify the best pretraining approach for a given dataset using a systematic search. Applied to multiple healthcare and retail datasets, we find consistent improvements over previously proposed pretraining approaches. In summary, the main contributions we provide are the following:

- PAITS introduces a novel combination of NLP-inspired pretext tasks and augmentations for self-supervised pretraining in irregularly sampled time series datasets.

- By leveraging a random search, PAITS consistently identifies effective pretraining strategies, leading to improvements in performance over previous approaches across healthcare and retail datasets.

## 2 Related Work

### 2.0.1 Self-supervised pretraining in other domains

In computer vision (CV) and natural language processing (NLP), where vast quantities of unlabeled data are available, researchers have explored an array of approaches for leveraging self-supervised learning, where pseudo-labels are automatically generated and used for pretraining models before finetuning with smaller labeled datasets. In CV, the current state of the art approaches involve contrastive learning, in which models are encouraged to be invariant to differently-augmented versions of the same input image. Thus, such methods, such as MoCo He et al. (2020) and SimCLR Chen et al. (2020), rely on the choice of image-related augmentations, such as color jitter and rotation. In NLP, where text data is represented by sequences of tokens, recent architectures such as transformers Vaswani et al. (2017), paired with self-supervised pretraining with large unlabeled datasets, have led to great improvements in NLP tasks by capturing contextual information about elements in a sequence given the other elements. In particular, two of the most common pretraining tasks are (1) "language modeling": predicting the next element given the previous elements of the sequence Peters et al. (2018); Radford & Narasimhan (2018), and (2) "masked language modeling": masking elements of a sequence and predicting the masked elements from the unmasked ones Devlin et al. (2018).

### 2.0.2 Self-supervised pretraining for time series data

Inspired by progress in CV and NLP in recent years, researchers have also begun to adapt self-supervised pretraining approaches for time series data. For example, several methods developed for dense time series data (particularly signal data that follow cyclical patterns, such as brainwaves and electricity usage), have involved contrastive learning-based approaches. The approaches introduced for time series data have often relied on new augmentations that reflect invariances expected specifically for time series data. For example, the TS-TCC method involves augmentations such as adding random jitter to the signals, scaling magnitudes within a specific feature, and shuffling chunks of signals (which is most appropriate for repeating patterns) Eldele et al. (2021). Similarly, TF-C incorporates the frequency domain for its contrastive learning approach Zhang et al. (2022). While these approaches have shown promise, they rely on an underlying expectation dense and regularly sampled time series data (often with perceptible repeating patterns). In practice, when data are sparse and irregularly sampled (e.g., Appendix Table 4), the use of such methods is limited by extreme missingness in the discretized representation and need for imputing the vast majority of data.

Beyond the more common contrastive learning-based methods, another recent pre-training approach by Zerveas et al. (2021) introduced an input denoising pretext task for regularly sampled time series data inspired by advances in NLP. In particular, they masked values for randomly sampled times and features, and as a self-supervised pretraining task, reconstructed the masked values, which led to improved downstream classification performance.

### 2.0.3 Data augmentations and augmentation selection

While pretraining tasks often involve the use of augmentations (e.g., masking parts of an input in order for them to be reconstructed as a task), the use of augmentation alone has been explored as an approach for improving the stability and generalizability of representations in supervised training regimes. For example, in computer vision, augmentation methods are routinely used in training neural networks for supervised tasks in a variety of domains Krizhevsky et al. (2012); He et al. (2016); Esteva et al. (2017). Similar approaches have also been explored in time series data, ranging in complexity from simple noise addition and scaling to complex neural network-based approaches such as generative adversarial networks Harada et al. (2018); Lou et al. (2018).

However, as found in CV, choosing augmentations is not always straightforward, and the choice of appropriate augmentation may vary across datasets and settings. While some work has explored complex procedures for selecting augmentation strategies (e.g., reinforcement learning-based Cubuk et al. (2019)), recent work has demonstrated that randomly selecting augmentations can sometimes perform equally well Cubuk et al. (2020). In our work, we explore the use of multiple imputation strategies, and employ a random search to identify an appropriate solutions for datasets.

### 2.0.4 Irregularly sampled time series data: Alternative data representations and pretraining

When time series data are sparse and irregularly sampled, the traditional approach of representing these data as discretized matrices may become impractical due to high sparsity, and thus imputation that is be required (Figure 1). These challenges have motivated the recent use of a sequence-based representation Horn et al. (2020); Tipirneni & Reddy (2022), in which time series data are encoded by a sequence of observed events, with parallels to NLP's representation of text data as a sequnce of tokens. In this context, inspiration may be drawn from self-supervised pretraining advances in NLP. For example, STraTS uses a sequence-based representation of time series, and introduces a forecasting pretraining task (similar to language modeling) where the goal is to predict feature values in a future time point given an input sequence of observations Tipirneni & Reddy (2022).

In this work, we draw on inspiration from NLP pretraining tasks, existing augmentation methods, and the previously proposed sequence-based representations of time series data. We uniquely combine pretext tasks and augmentations in the context of irregularly sampled time series data and present a systematic search for appropriate pretraining strategies multiple datasets and domains.

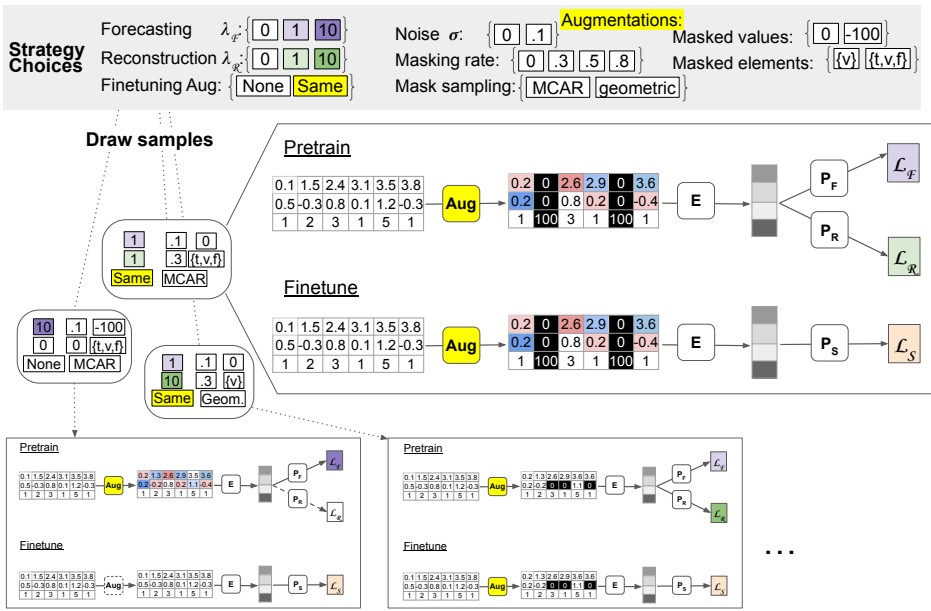

Figure 2: PAITS illustration: We sample and select pretraining and augmentation strategies (details in Appendix Algorithm 1).

# 3  Methods

## 3.1  Notation and data representation

As illustrated in Figure 1, each time series is represented by a set of observation triplets which each encode a feature, time it was measured, and observed value at that time. We use a similar data representation to the one proposed by Tipirneni & Reddy (2022), which we describe below:

### 3.1.1  Labeled dataset notation

Consider a labeled dataset of $N_L$ samples: $\mathcal{D}_\mathcal{L} = \{(\mathbf{d}_i, \mathbf{S}_i, \mathbf{y}_i)\}_{i=1}^{N_L}$. For each sample $i$, we have static features $(\mathbf{d}_i)$ that do not vary with time, a time series $(\mathbf{S}_i)$ which we define next, and an associated label $(\mathbf{y}_i)$. More specifically, the time series for instance $i$, where $M_i$ observations occurred within a given observation window, is given by $\mathbf{S}_i = (s_1, ... s_{M_i})$, where each observation $s_j$ is represented by a triplet $(t_j, v_j, f_j)$, where $f_j \in 1, ... V$ represents which feature was observed (among $V$ total available features), $t_j$ is the time that the feature was measured, and $v_j \in \mathbb{R}$ is the observed value of the feature $f_j$ at time $t_j$. Often, $S_i$ may be restricted to observations occurring in a specific period from which predictions will be made (e.g., the first 48 hours of a hospital stay). Given $\mathbf{d}_i$ and $\mathbf{S}_i$, the goal is to predict the accompanying label $\mathbf{y}_i$.

### 3.1.2  "Unlabeled" dataset for self-supervised pretraining

For self-supervised pretraining, we consider an additional set of available data (which may overlap with the labeled dataset, but does not require labels $y$). We have an "unlabeled" dataset with $N_U$ (where usually $N_U \geq N_L$) samples: $\mathcal{D}_\mathcal{U} = \{(\mathbf{d}_i, \mathbf{S}_i)\}_{i=1}^{N_U}$. Here, each time series $\mathbf{S}_i$ may contain additional observations outside of the times expected for the supervised task (e.g., an entire hospital stay, rather than the first 48 hours), and we do not expect access to a label. As described in the following sections, we generate pretraining input data along with pseudo-labels for our pretext tasks from unlabeled time series samples $\mathbf{S}_i$.

From our unlabeled dataset $\mathcal{D}_\mathcal{U}$, we generate input data and pseudo-labels for our pretext tasks. In particular, we define observation length $(l_o)$, forecasting length $(l_f)$, and stride length $(l_s)$ which we use to generate a larger set of processed samples used for pretraining. We then define a collection of starting points, $W$, for observation windows beginning at the first possible start-time for the data, and then at at intervals of

the stride length ($W = \{0, s, 2s, ...\}$. For a given time series sample $S_i$ and observation start time $t_w$, we thus have a pretraining input sample $S_{iw} = \{(t_j, v_j, f_j) \in S_i : t_w <= t_j < t_w + l_o\}$. We note that unlike with a discretized representation for regularly sampled time series, elements in $S_{iw}$ are not regularly spaced observations from time $t_w$ to $t_w + l_o$; instead, we have a variable-length sequence of triplets depending on the true number of observations that occurred during that time period.

### 3.2 Self-supervised pretraining and finetuning tasks

Consistent with Tipirneni & Reddy (2022), we use a neural network architecture in which these triplets are passed through learned embedding layers and a transformer to generate a low-dimensional representation of the time series. Built upon this base encoder architecture, we add additional layers to encode two pretext tasks and a final target task which we describe in the following subsections.

***Forecasting pretraining.*** The first of our pre-text tasks, which was originally proposed by Tipirneni & Reddy (2022), is a forecasting task, in which for each sample, we predict the observed value of each feature in a pre-determined follow-up window, based on the events from an observation window. We thus have a $V$-dimensional regression task. In this work, we propose incorporating augmentations to the input data, which we describe in the following subsections sections.

Due to the irregularly sampled nature of the time series data, a feature may be observed multiple times during the follow-up window, or more commonly, not have been observed at all in the follow-up window. Thus, for a given sample $i$ with observation window start $t_w$, our inputs are $\mathbf{d}_i$ and $\mathbf{S}'_{iw}$ (an augmented version of $\mathbf{S}_{iw}$), and the goal is to predict the first observed value of each feature (if such a value exists). Thus, our masked mean squared error loss (such that predictions for missing terms are ignored) is given by:

$$\mathcal{L}_\mathcal{F} = \frac{1}{N_U} \sum_{i=1}^{N_U} \sum_{w \in W} \sum_{j=1}^{V} m_{iw,j} (\hat{z}_{iw,j} - z_{iw,j})^2$$

where $\hat{z}_{iw}$ is the model's prediction vector for sample $i$ with an observation window starting at $w$, $z_{iw}$ is the true $V$-dimensional array of observed values in the follow-up window, and $m_{iw,j}$ is a mask with value 1 if feature $j$ was observed in the follow-up window for sample $i$, and 0 otherwise.

***Reconstruction pretraining.*** A second pretext task which we propose to add, inspired by masked language modeling, is to reconstruct the original time series $S_{iw}$ from augmented inputs $S'_{iw}$. In particular, for most of our datasets, consistent with the forecasting task, we predict the values observed in the original time series. To that end, we include a decoder module which takes the contextual triplet embeddings produced by the transformer layers and generates a *seqlen*-dimensional output indicating predictions of the original value associated with each event. The loss for reconstructing time series $S_{iw}$ is given by:

$$\mathcal{L}_\mathcal{R} = \frac{1}{N_U} \sum_{i=1}^{N_U} \sum_{w \in W} \sum_{k=1}^{seqlen} p_{iw,k} c_{iw,k} (\hat{v}_{iw,k} - v_{iw,k})^2$$

where $p_{iw} \in \{0, 1\}^{seqlen}$ is a padding mask indicating whether the index $k$ represents a real observation (1) or padding (0), and $c_{iw} \in \{0, 1\}^{seqlen}$ is a reconstruction mask indicating whether the element should be reconstructed. An all 1s vector would indicate that the goal is to reconstruct the entire sequence, whereas alternative masks (which we describe in relation to augmentations below) can be used to encode the goal of reconstructing only parts of the sequence.

For pretraining, we jointly optimize for both pretext tasks, and allow different weightings of the tasks (e.g., Appendix Table 5), given by hyperparameters $\lambda_\mathcal{F}$ and $\lambda_\mathcal{R}$:

$$\mathcal{L}_\mathcal{P} = \lambda_\mathcal{F} \mathcal{L}_\mathcal{F} + \lambda_\mathcal{R} \mathcal{L}_\mathcal{R}$$

***Supervised finetuning.*** After convergence on the jointly pretrained tasks above, we discard the pretraining-specific modules and fine-tune the model on the target task of predicting $\mathbf{y}_i$ from $\mathbf{d}_i$ and $\mathbf{S}_i$. The prediction layers for this final target are built on the base encoder described above (which has been optimized to

the pretext tasks). Thus, the final model is initialized with the pretrained encoder weights (followed by randomly initialized prediction layers), and during finetuning, model parameters are updated to minimize the supervised task's loss $\mathcal{L}_{\mathcal{S}}$ (e.g., cross-entropy loss for a classification task). We illustrate the full pretraining and finetuning process in Figure 2.

### 3.3 Augmentations

The goal of our self-supervised pre-training strategy is to generate robust representations that reflect natural variations in irregularly sampled data (e.g., dropped observations) as well as general variation in real-world data (e.g., measurement noise). We expect that invariance to such perturbations may lead to more effective downstream classification. To that end, we propose combining two classes of augmentations (which we illustrate in toy examples in Figure 2):

***Adding noise***. In particular, we augment samples by applying Gaussian noise to the continuous-value elements of the triplets: time and value. Here, $\sigma \in \mathbb{R}_{\geq 0}$ is a hyperparameter controlling the amount of Gaussian noise used to perturb the true values, and our augmented view for sample $i$, $s_i' = (t_i', v_i', f_i)$ is given by:

$$t_{i,j}' = t_{i,j} + \epsilon_{i,j}, \text{ where } \epsilon_{i,j} \sim N(0, \sigma^2)$$

$$v_{i,j}' = v_{i,j} + \epsilon_{i,j}, \text{ where } \epsilon_{i,j} \sim N(0, \sigma^2)$$

where each observation $j$ is independently perturbed across sample $i$'s time series. In our experiments, we also consider $\sigma := 0$, implying no augmentation to the input.

***Masking***. Here, because our data represents sparse and irregularly sampled time series, we apply additional augmentations to further down-sample the time series. For the masking augmentation, we consider several hyperparameters to cover a wide range of datasets and scenarios: mask rate (probability of a specific observation being masked), mask sampling strategy (probability distribution for sampling missingness sequences), which parts of the triplet to mask, and the masked values themselves.

First, we consider the mask rate $r$ and mask sampling strategy, which together determine the probability distributions governing sampled binary masks. For mask sampling strategy we consider (1) simply sampling from a Bernoulli distribution where each observation across each time series is randomly masked with probability $r$, and (2) sampling masks that follow a geometric distribution, as proposed by Zerveas et al. (2021). In particular, for the approach described by Zerveas et al. for regularly sampled time series, state transition probabilities from masked to unmasked and vice versa follow geometric distributions, leading to longer alternating stretches of masked vs. unmasked observations. We adapt their approach by selecting time increments at which to mask, such that masks are consistently applied within a given interval (described in depth in the Appendix).

Given a sampled binary mask vector $m_i$ for time series $s_i$ as described above, we can then generate augmented (masked) versions $t_i, v_i, f_i$ as follows:

$$t_{i,j}' = t_{i,j} m_{i,j} + a_t(1 - m_{i,j})$$

$$v_{i,j}' = v_{i,j} m_{i,j} + a_v(1 - m_{i,j})$$

$$f_{i,j}' = f_{i,j} m_{i,j} + a_f(1 - m_{i,j})$$

where $a = (a_t, a_v, a_f)$ is the masked value with which we replace the original values. Finally, we consider masking different portions of the triplets within each time series, so our final masked series $s_i'$ is:

$$s_i' = (\mathbb{1}_{t \in E} t_i' + \mathbb{1}_{t \notin E} t_i, \mathbb{1}_{v \in E} v_i' + \mathbb{1}_{v \notin E} v_i, \mathbb{1}_{f \in E} f_i' + \mathbb{1}_{f \notin E} f_i)$$

where $E$ is the set of elements within the triplet that will be masked. We apply the augmentations one after another (noise $\rightarrow$ masking) and note that these augmentations are used during pretraining and then optionally also used during finetuning, as shown in FIgure 2 and Appendix Table 5.

### 3.4 PAITS Strategy Search

We hypothesize that datasets with varying distributions and sparsity patterns will benefit from different pretraining approaches; thus, we propose using a random search strategy to explore the space of possible pretraining tasks and augmentations, which we outline in Appendix Algorithm 1. In Figure 2 and Appendix Table 5, we summarize and illustrate the pretraining and finetuning search space we considered.

Finally, to identify an appropriate pretraining and finetuning strategy within each dataset, we randomly sample strategies from the search space defined in Appendix Table 5, perform pretraining and finetuning on the training sets (described in the next section), and select the strategy with the best validation performance obtained during finetuning.

## 4 Experimental Settings

### 4.1 Datasets and pre-processing

We apply PAITS to four datasets with sparse and irregularly sampled time series: three commonly used real-world medical datasets (with a goal of predicting in-hospital mortality), and a retail dataset (with the aim of predicting purchases). We summarize them below and in Appendix Table 4:

**PhysioNet 2012:** The PhysioNet Challenge 2012 dataset Silva et al. (2012) is a publicly available dataset of patients in intensive care units (ICUs). Each patient's sample includes 48 hours of time series data, from which the goal is to predict in-hopsital death.

**MIMIC III:** The MIMIC III dataset Johnson et al. (2016) provides a large collection of medical records from Beth Israel Deaconess Medical Center ICU stays, and we similarly perform 48h mortality prediction. However, for this dataset, we have access to longer time series beyond the first 48h, which we leverage as unlabled pretraining data.

**eICU:** The eICU Collaborative Research Database Pollard et al. (2018) is a large multi-center database consisting data from ICU stays, for which the primary task is to predict mortality from the first 24h of data, although additional time series data are available for pretraining.

**H&M:** We use the data from the "H&M Personalized Fashion Recommendations" competition hosted on Kaggle, which consists of purchases made over the course of two years by H&M customers, ranging from September 2018 to September 2020, with the goal of predicting a user's purchases based on past purchases.

Further details about data are provided in the Appendix.

#### 4.1.1 Medical data preprocessing

Consistent with the approach used by Tipirneni & Reddy (2022), and described above, we represent time series data as a sequence of times, features, and values. Our labeled datasets consist of samples with time series data available for the first 24- or 48-hours in the ICU, along with binary mortality labels, which we randomly split into training, validation, and test splits (described further in the Appendix). When available, we generate a larger unlabeled dataset consisting of time series data outside of the supervised task windows, as described earlier.

For stability and efficiency of neural network training, we normalize times and values (within features) to have mean of 0 and variance of 1, and set a maximum sequence length for each dataset as the 99th percentile across samples (Appendix Table 4). Additional details may be found in the Appendix.

#### 4.1.2 Retail data preprocessing

To evaluate our pretraining strategies on an alternative data type with even higher sparsity, we apply the same approach for the medical datasets with slight modifications for the H&M dataset. We restrict our focus to customers with at least 50 purchases throughout the full two year period (top 12.5% of customers), and items that were purchased at least 1000 times (7804 items). For our time series representations, each

| Dataset (Approx. training dataset size: labeled / unlabeled) | Methods | Labeled data (%) | | | |
|---|---|---|---|---|---|
| | | 10% | 20% | 50% | 100% |
| **PhysioNet 2012** (6.4K / 6.2K) | STraTS | 0.4097±0.0279 | **0.4460±0.0128** | 0.4735±0.0132 | 0.5019±0.0046 |
| | TST | 0.2871±0.0332 | 0.3433±0.0486 | 0.4411±0.0106 | 0.4818±0.0064 |
| | TS-TCC | 0.3076±0.0222 | 0.3709±0.0368 | 0.4504±0.0076 | 0.4961±0.0077 |
| | CL (PAITS augs) | 0.3191±0.0098 | 0.3436±0.0271 | 0.4509±0.0147 | 0.4879±0.0034 |
| | No pretraining | 0.2787±0.0289 | 0.3472±0.0430 | 0.4356±0.0162 | 0.4762±0.0103 |
| | PAITS | **0.4201±0.0213** | 0.4422±0.0150 | **0.4862±0.0111** | **0.5104±0.0046** |
| **MIMIC III** (29K / 422K) | STraTS | 0.5170±0.0111 | 0.5469±0.0136 | 0.5801±0.0069 | 0.5872±0.0034 |
| | TST | 0.4751±0.0255 | 0.5076±0.0129 | 0.5505±0.0107 | 0.5655±0.0136 |
| | TS-TCC | 0.5342±0.0181 | 0.5487±0.0066 | 0.5652±0.0068 | 0.5768±0.0037 |
| | CL (PAITS augs) | 0.4089±0.0212 | 0.4658±0.0105 | 0.5236±0.0130 | 0.5574±0.0073 |
| | No pretraining | 0.4737±0.0176 | 0.5111±0.0136 | 0.5424±0.0093 | 0.5665±0.0037 |
| | PAITS | **0.5394±0.0177** | **0.5632±0.0083** | **0.5868±0.0094** | **0.5975±0.0088** |
| **EICU** (85K / 1.07M) | STraTS | **0.3288±0.0084** | 0.3356±0.0143 | **0.3528±0.0043** | **0.3639±0.0049** |
| | TST | 0.2650±0.0200 | 0.3072±0.0119 | 0.3339±0.0058 | 0.3520±0.0050 |
| | TS-TCC | 0.3116±0.0072 | 0.3294±0.0083 | 0.3427±0.0052 | 0.3610±0.0044 |
| | CL (PAITS augs) | 0.2986±0.0118 | 0.3196±0.0075 | 0.3375±0.0052 | 0.3528±0.0042 |
| | No pretraining | 0.2716±0.0175 | 0.2961±0.0182 | 0.3244±0.0071 | 0.3462±0.0041 |
| | PAITS | 0.3280±0.0199 | **0.3380±0.0078** | 0.3523±0.0073 | **0.3603±0.0113** |

Table 1: After pretraining and finetuning, we compare test AUROCs across methods for three healthcare datasets. We provide additional details about datasets and test AUPRC metrics in Appendix Tables 4 and 6, respectively.

triplet $(t, f, v)$ in the sequence of events represents a date, price, and article ID for an item that the customer purchased.

We considered our supervised task to be predicting users' purchases in September 2020 from their purchases in August 2020, but leverage the earlier months' data for generating a much larger pre-training dataset. As shown in Appendix Table 4, we note that these time series are extremely sparse: on average, there is only about one purchase in a given month.

## 4.2 PAITS implementation details

For our healthcare data experiments, we use the encoder architecture proposed by Tipirneni & Reddy (2022), which take the time series and demographic features as input, and consists of the following components: (1) Separate embedding modules for time, values, and features, from which the three embeddings are summed to obtain a final triplet embeddings for each event, (2) a standard transformer architecture Vaswani et al. (2017) to incorporate contextual information across triplet embeddings, (3) a fusion self-attention layer that generates a weighted average over the triplets' embeddings to produce a final embedding over the entire time series, and (4) a static features embedding module, from which the time series and static features embedding are concatenated to obtain the final embedding (more details in Appendix).

Built on the encoder architecture, we have three task-specific modules: (1) the forecasting module, consisting of a single dense layer used for the pretraining forecasting task, (2) a reconstuction module, consisting of three dense layers used for the reconstruction pretraining task, and finally (3) a prediction module consisting of two dense layers for the final supervised task. While the architecture is flexible, we held it constant for our PAITS strategy search experiments for simpler comparison across datasets and methods. For the strategy search, we considered pretraining, augmentation, and finetuning settings outlined in Appendix Table 5, and sampled 100 distinct strategies in our search.

| Dataset | $(\lambda_{\mathcal{F}}, \lambda_{\mathcal{R}})$ | Aug. noise | Mask sampling, rate | Mask elements → values | Finetuning aug. |
|---|---|---|---|---|---|
| MIMIC III | (1,1) | 0 | random, 0.5 | (t,f,v)->(0,0,V+1) | None |
| PhysioNet2012 | (1,0) | 0.1 | geometric, 0.5 | (t,f,v)->(0,0,V+1) | Same |
| EICU | (1,1) | 0 | geometric, 0.8 | (t,f,v)->(-100,-100,V+1) | None |
| H&M | (1,0) | 0.1 | random, 0.3 | (t,f,v)->(-100,-100,V+1) | Same |

Table 2: Strategies for pretraining, augmentation, and finetuning selected by PAITS across datasets.

| Methods | Labeled data (%) | | | |
|---|---|---|---|---|
| | 10% | 20% | 50% | 100% |
| STraTS | 0.0147±0.0005 | 0.0152±0.0003 | 0.0153±0.0004 | 0.0157±0.0003 |
| TST | 0.0129±0.0004 | 0.0132±0.0002 | 0.0133±0.0001 | 0.0134±0.0002 |
| TST (random mask) | 0.0130±0.0003 | 0.0132±0.0002 | 0.0133±0.0001 | 0.0131±0.0003 |
| CL (PAITS augs) | 0.0147±0.0003 | 0.0148±0.0003 | 0.0152±0.0004 | 0.0151±0.0008 |
| No pretraining | 0.0130±0.0003 | 0.0132±0.0002 | 0.0131±0.0005 | 0.0132±0.0003 |
| PAITS | **0.0148±0.0006** | **0.0154±0.0002** | **0.0158±0.0006** | **0.0161±0.0003** |

Table 3: After pretraining and finetuning, we compare purchase prediction effectiveness across methods in the H&M dataset. In Appendix Table 7, we additionally provide test binary cross-entropy loss values.

### 4.3 Baselines

We compare our approach to related methods for time series pretraining, and provide details in the Appendix:

- **STraTS**: A related approach developed for irregularly sampled time series data, with the same data representation and base architecture Tipirneni & Reddy (2022). STraTS represents one strategy within our search space: forecasting alone ($\lambda_{\mathcal{R}} = 0$) and no augmentations.

- **TST**: This reconstruction pretraining method was developed for regularly sampled time series Zerveas et al. (2021), and we modify their approach for our data representation. This approach is similar to masked value reconstruction alone ($\lambda_{\mathcal{F}} = 0$) with geometric masking.

- **TS-TCC**: A contrastive learning-based pretraining approach Eldele et al. (2021) which we have adapted to our data representation. TS-TCC learns joint forecasting and contrastive tasks, alongside augmentations including scaling, noise, and permuting time blocks.

- **Contrastive learning with PAITS augmentations**: Here, we consider the same set of augmentations used in PAITS; however, we replace the the reconstruction and forecasting tasks with a contrastive task (i.e., InfoNCE loss Oord et al. (2018)) for the strategy search.

- **No pretraining**: Random initialization to show the minimum performance expected without any pretraining.

## 5 Experimental Results and Discussion

In this section, we evaluate PAITS alongside alternative pretraining approaches across multiple datasets and goals.

### 5.0.1 ICU mortality prediction

As described above, our ultimate goal for the healthcare dataets is mortality prediction based on a patient's initial stay in the ICU. Separately for each dataset, we apply our strategy search approach, and sample 100 strategies for pre-training and fine-tuning as outlined in Appendix Table 5. Within each run, pretraining is done with a larger unlabeled dataset containing data collected from the first five days in the ICU. After convergence, finetuning is applied to only the labeled samples from the initial 24h or 48h window. We select the strategy for which the performance is best among the validation set.

In Table 1, we report test results of PAITS and related methods. To demonstrate the effectiveness of our approach when labeled datasets are smaller, which is often the motivation for leveraging self-supervised pretraining, we provide test metrics when we vary the amount of labaled data available during finetuning (while keeping pretraining constant).

As shown in Table 1, PAITS systematically finds combinations of pretraining tasks and augmentations, improving prediction accuracy compared to previous methods. We also note that differences among methods

tend to be more pronounced as labeled dataset sizes decrease, highlighting the advantage of more robust pretraining in the smaller data regime. Furthermore, we note that the relative performance of baseline methods varies across datasets, highlighting the need for a systematic search among candidate pretraining approaches. Indeed, in Table 2, we show that different strategies were selected for each dataset, possibly due to underlying differences in their distributions.

One notable observation was that when comparing the two pretext tasks we considered in our strategy search, forecasting tended to provide more gain than reconstruction. In fact, for PhysioNet 2012, where there were no additional unlabeled samples available for preraining (i.e., we pretrain and finetune with the same samples), we found that forecasting alone was selected by PAITS. This may indicate that the reconstruction task relies on larger sets of unlabeled data to provide robust improvements to the pretrained representations, and thus was more helpful in the larger MIMIC III and eICU datasets.

### 5.0.2 Retail purchase prediction

Next, to evaluate our approach on a different domain, we consider the problem of forecasting a user's purchases based on their prior purchases using the H&M dataset described above. We leverage the PAITS method in largely the same way as with healthcare data with some modifications for the specific retail forecasting task: (1) Rather than binary classification (mortality), our supervised task is multilabel classification (i.e., for each article, whether or not it was purchased in the prediction window), and (2) for both reconstruction and forecasting pretext tasks, we consider the feature (i.e., purchased item) as the key element of the triplet, and thus we use binary cross-entropy loss for each article (rather than mean squared error when predicting values in the healthcare datasets).

As described in Methods, we leverage the large set of previous months' purchases for pretraining, and set a final goal of predicting September 2020 purchases from August 2020 purchases. Consistent with the evaluation metric in the Kaggle competition, we use a MAP@12 metric (but over the course of one month) to evaluate the relative rankings of predicted articles purchased. As shown in Table 3, PAITS is able to identify a pretraining and finetuning strategy that most effectively predicts purchased items in the following month. Interestingly, similarly to the PhysioNet 2012 dataset above, we also found here that the forecasting task without reconstruction was selected by our search. This could be due to the forecasting pretraining task being identical to the supervised task in this regime, which may highlight the importance of alignment between pretraining and finetuning tasks. However, the additional augmentations selected by PAITS lead to improvments over the non-augmented forecasting introduced by STraTS, highlighting the utility of considering a range of tasks and augmentations.

## 6 Conclusion

In this work, we present PAITS, a systematic approach for identifying appropriate pretraining and finetuning strategies for sparse and irregularly sampled time series data. We found that different datasets do indeed benefit from different combinations of pretext tasks, alongside different augmentation types and strengths, even when downstream tasks are similar. Thus, a fixed strategy may not always be relied on for consistent gains during pretraining. Furthermore, we found that the use of NLP-inspired pretexts tasks for the sequence-based representation of time series data was more effective than a contrastive learning pretext task, which has been more effective in the context of dense and regularly sampled time series data. While PAITS was developed with the goal of improving pretraining for irregularly sampled time series, such a framework could be similarly applied to dense and regularly sampled time series; however, future work will be needed to assess whether similar gains are seen in that regime. Finally, while our strategy search was restricted to a limited set of pretraining tasks and augmentations, the approach could be arbitrarily extended to include a wider variety of tasks and augmentations as they are developed, which may open the door to even greater gains in prediction for irregularly sampled time series data.

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
