# OpenReview forum: "PAITS: Pretraining and Augmentation for Irregularly-Sampled Time Series"
_TMLR — Withdrawn by Authors_

### Review · Reviewer_p6Bj · 2024-04-29

**Summary Of Contributions:**

This work focuses on pretraining strategies for irregularly sampled and sparse time-series data, a challenging problem in the time-series analysis field. This work is motivated by the fact that existing pretraining and augmentation strategies do not necessarily focus on irregular time series.

This work proposes a new method called PAITS that searches for the best combination of pretraining and augmentation techniques for self-supervised pretraining on time-series data.

**Audience:**

Yes

**Claims And Evidence:**

No

**Requested Changes:**

See the review comments.

**Strengths And Weaknesses:**

Strength:

- This work is motivated for irregular time-series data, which is an interesting area of work as regular sampling for time-series data is a common challenge in real-world applications.

- This work extends simply proposing a one-size-fits-all solution to accommodate the diversity in time-series datasets.

- This work conducts a diverse empirical analysis to showcase the performance of PAITS.

Weaknesses:

- This work's main motivation is pretraining approaches for irregular time-series data. However, the proposed algorithm does not consider the sample irregularity in its search algorithm. Or at least it is unclear how irregularly sampled data is approached in this method. The suggested pretext pretraining is designed to be a general method that can work on irregular time-series dataset, and not to consider the limitations of irregular samples [R1]. The motivation for this purpose should be clarified.

- In addition to the previous point, this work does not consider the synchronization challenge of the observation time-steps of irregular time-series. There is no specific reason why PAITS is motivated specifically for irregular time-series data.

- There is a marginal empirical improvement shown in the conducted experiments, essentially in Table 1. According to the motivation of this paper, one would expect better results for a low-label regime. The impact of PAITS should be further investigated.

- PAITS is described as a random search process that picks the right algorithms for pretraining/augmentation based on the validation performance. This is a naive approach as the algorithms considered in pretraining/augmentation are not novel or designed for irregular time series. A more in-depth analysis should be provided to correlate the characteristics of each dataset to the combination of the algorithm chosen to pre-train the model on it. There is a lack of theoretical or empirical insights into the choice of the right pretraining strategy.

[R1]: Chen, Yuqi, et al. "ContiFormer: Continuous-time transformer for irregular time series modeling." Advances in Neural Information Processing Systems 36 (2024).

---

### Review · Reviewer_ayUw · 2024-05-06

**Summary Of Contributions:**

The paper introduces PAITS (Pretraining and Augmentation for Irregularly-sampled Time Series), a  framework designed to optimize pretraining strategies for irregularly sampled and sparse time series data. The techniques are mainly inspired by the pretraining methods widely used in NLP. Some empirical studies have been performed to support their claims.

**Audience:**

Yes

**Broader Impact Concerns:**

No broader impact concerns.

**Claims And Evidence:**

No

**Requested Changes:**

1. [Critical] Add necessary discussions on the algorithm's complexity of the random search algorithm and the computational cost in the presented tasks.

2. [Critical] Add necessary ablation studies to provide a clearer view of the effectiveness of the proposed techniques.

3. [Minor] Move important details from the appendix to the main text and fix formatting problems and typos.

**Strengths And Weaknesses:**

## Strengths
1. I am not an expert in this area. However, it is interesting to see the applications of the techniques that originated from the NLP tasks in other time-series tasks.

## Weakness
1. The use of a random search algorithm, while effective, might introduce significant computational overhead, especially with very large datasets, which could limit its applicability in resource-constrained environments. I think the authors could provide more details on the algorithm's complexity and what is the computational cost in the presented tasks.

2. While there is a list of techniques proposed, the lack of ablation studies makes it difficult to see their effectiveness. Besides, there is no clear guidance on how to apply them according to the properties of the time series data.

3. Many important details should have been included in the main text instead of the appendix.

4. There are also some obvious formatting problems and typos.

---

### Review · Reviewer_ZPYi · 2024-05-22

**Summary Of Contributions:**

This paper propose a framework for identifying suitable pretraining strategies for sparse and irregularly sampled time series datasets. Specifically, the proposed PAITS method combines forcasting and reconstruction training objectives with multiple data augmentation options, and employs a random search approach to find the most suitable design of the pretraining and finetuning strategy given the dataset.

**Audience:**

Yes

**Broader Impact Concerns:**

No broader impact concern.

**Claims And Evidence:**

Yes

**Requested Changes:**

More discussion is needed on the impact of random search: How is the variance between the performance of different choices? How many trials are needed to find the optimal result? Whether this random selection is scalable to a larger search space?

**Strengths And Weaknesses:**

## Strength
1. This paper focuses on an important problem of pretraining models for sparse and irregularly sampled time series. The paper provides a clear introduction on the importance of such data type and the difficulty it induces for the pretraining.
2. The paper identifies the impact of different training objective and data augmentation strategies on the final outcome of the pretraining, and that such impact is different for different dataset. The idea of searching through multiple combinations is helpful in finding the best setting.

## Weakness
1. The pretraining method description is not clear, from the current formulation it is hard to tell if the pretraining is done by using all the observed features to forcast or reconstruct unseen features, or only the selected feature is used to predict itself. A figure illustrating the input and target of the pretraining objective will be helpful.
2. Performance-wise, seems like the performance improvement brought by the proposed method diminishes with a larger unlabeled dataset., such as on the EICU dataset. Further analysis is needed on whether this incapability is caused by the proposed pretraining objective or by the difficulty of finding an optimal option with trandom search for larger dataset.
3. The impact of random search to the result is not well-studied. It is unclear about the cost of the search, like how many trials are needed. It is also unclear about the impact of different options on the final performance. Further ablation study is needed to verify whether the random search is finding an optimal result.

---

### Note · Authors · 2024-06-05

**Comment:**

We would like to thank the editor in chief and all the reviewers. We have decided to withdraw the paper and improve it further.

**Withdrawal Confirmation:**

I have read and agree with the venue's withdrawal policy on behalf of myself and my co-authors.